# Proof-of-Concept Human Organ-on-Chip Study: First Step of Platform to Assess Neuro-Immunological Communication Involved in Inflammatory Bowel Diseases

**DOI:** 10.3390/ijms241310568

**Published:** 2023-06-24

**Authors:** Tristan Gabriel-Segard, Jessica Rontard, Louise Miny, Louise Dubuisson, Aurélie Batut, Delphine Debis, Mélanie Gleyzes, Fabien François, Florian Larramendy, Alessandra Soriano, Thibault Honegger, Stéphane Paul

**Affiliations:** 1CIRI—Centre International de Recherche en Infectiologie, Team GIMAP, Université Claude Bernard Lyon 1, Inserm, U1111, CNRS, UMR530, CIC 1408 Vaccinology, 42023 Saint-Etienne, France; 2Service de Psychiatrie Transversale, Centre Hospitalo-Universitaire de Saint Etienne, Hôpital Nord, 42055 Saint-Etienne, France; 3NETRI, 69007 Lyon, France; 4Internal Medicine Department, Gastroenterology Division and IBD Center, Azienda Unità Sanitaria Locale—IRCCS, 42122 Reggio Emilia, Italy; alessandra.soriano@ausl.re.it

**Keywords:** inflammatory bowel diseases, gut-brain axis, neuroimmunology, microfluidic technology, electrophysiological activity, human induced pluripotent stem cells

## Abstract

Inflammatory bowel diseases (IBD) are complex chronic inflammatory disorders of the gastrointestinal (GI) tract. Recent evidence suggests that the gut-brain axis may be pivotal in gastrointestinal and neurological diseases, especially IBD. Here, we present the first proof of concept for a microfluidic technology to model bilateral neuro-immunological communication. We designed a device composed of three compartments with an asymmetric channel that allows the isolation of soma and neurites thanks to microchannels and creates an in vitro synaptic compartment. Human-induced pluripotent stem cell-derived cortical glutamatergic neurons were maintained in soma compartments for up to 21 days. We performed a localized addition of dendritic cells (MoDCs) to either the soma or synaptic compartment. The microfluidic device was coupled with microelectrode arrays (MEAs) to assess the impact on the electrophysiological activity of neurons while adding dendritic cells. Our data highlight that an electrophysiologic signal is transmitted between two compartments of glutamatergic neurons linked by synapses in a bottom-up way when soma is exposed to primed dendritic cells. In conclusion, our study authenticates communication between dendritic cells and neurons in inflammatory conditions such as IBD. This platform opens the way to complexification with gut components to reach a device for pharmacological compound screening by blocking the gut-brain axis at a mucosal level and may help patients.

## 1. Introduction

Inflammatory bowel diseases (IBD) are chronic inflammatory conditions affecting the digestive system, comprising two main distinctive entities: ulcerative colitis (UC) and Crohn’s disease (CD). IBD is characterized by intricate inflammation and microbiota dysbiosis [1]. IBD patients also manifest neurocognitive symptoms such as sleep disturbance, low mood, fatigue, hopelessness, and avolition. A high frequency of these symptoms in IBD patients was reported in the last meta-analysis (25.2% in CD and 24.0% in UC), and an odd ratio for depressive symptoms in IBD versus healthy controls was reported at 1.9 (CI 95% 1.4–2.7) [2]. The pathophysiology of neurological manifestations in IBD could mainly involve the immune system, representing a contributory mechanism to the gut-brain axis dysfunction [3]. The gut-brain axis is a bidirectional communication system modulated by neural, hormonal, metabolic, immunological, and microbial signals and plays a pivotal role in the pathogenesis of IBD [4]. The gut-brain communication passes through the sensitive vagal nerve fibers present in the mucosa, completing the one throughout the enteric nervous system (ENS) [5], to reach the nucleus, *Tractus solitarus*, at the central nervous system (CNS) level [4,6]. Furthermore, the ENS senses the microbiota information and modulates the neurophysiology of the gut by an extrinsic neuro sympathetic circuit localized in the brain [7]. The release of pro-inflammatory factors mediates this neuro-immune communication: interleukin (IL)-1β, IL-6, and tumor necrosis factor (TNF)-α released from epithelial and immune cells. Beyond this paracrine mechanism, evidence exists for cell-to-cell interaction between nerve fibers and immune cells. These interactions activate electrophysiological signals in neurons by activating the neurons’ signaling cascade involving Ca^2+^ and cAMP, activating intracellular kinases and generating an action potential, and lowering the threshold to trigger action potentials throughout cation channels [8]. Evidence from electrophysiology studies shows that synaptic connections also exist between sensory neurons and enteroendocrine cells of the intestinal epithelium, transmitting nutrient signals from the lumen through glutamate to mucosal neurons and secondary neurons [9,10]. This circuit had been reconstituted in vitro by the co-culture of sensory neurons and enteroendocrine cells [11].

It is hypothesized that chronic mucosal inflammatory signals and neuronal cell dysregulation could explain the psycho-behavioral manifestations by reaching the CNS and causing alterations in brain functions. An option for curing the neuropsychological symptoms of IBD could reside in the modulation of immunological communication to nerves within the gut-brain axis. Modeling this bilateral communication remains challenging due to its high level of biological complexity, the involvement of different cell types and physical communication pathways, and the lack of study models. Most animal studies use fecal microbiota transfer in germ-free mice or mice deprived of bacteria by antibiotic therapy [12]. To the best of our knowledge, there are not any models available to study the effect of IBD pathophysiology elements on nerve cells in the mucosa. Some studies involve epithelial cells and gut organoids but do not include nervous cells [13]. Organs-on-Chip (OoC) is an innovative technology that reproduces an in vitro structured and relevant neuronal network to human physiopathology and can be used to compartmentalize several cell types. Beaurivage and colleagues have introduced an innovative model that employs primary cells obtained from patients [14]. This model involves co-culturing the cells in a microfluidic device comprising two chambers of epithelial cells and macrophages derived from monocytes. This device is a valuable tool for exploring the impact of inflammatory mediators on the physiology of macrophages and epithelial cells. However, there seems to be no gut-on-chip model that incorporates nervous cells in a multi-chamber device [14]. Microfluidic devices with multiple chambers allow for cultivating various cell types in multiple compartments. This addresses our goal of modeling the impact of immune activation-induced inflammation on nerve cells. However, current gut-on-chip models provide histological and soluble molecule analysis that is not entirely applicable to the functional analysis of neurons. Compartmentalization is possible thanks to connection via microchannels, through which the neuronal extensions will pass while evaluating the electrophysiological impact [15].

In this study, we propose a microfluidic platform for modeling the immuno-nerve pathway of the gut-brain axis, which aims to develop an in vitro platform to study neuron-immune cell communication. The microfluidic device comprises three compartments with an asymmetric channel, allowing the isolation of soma and neurites thanks to microchannels to create in vitro synaptic transmission. Each channel is fluidically isolated thanks to the microchannel architecture (3 µm in height and 10 µm in width), allowing neurites and axons to project into it. The microfluidic device was coupled with MEA to record the electrophysiological signal according to the conditions and analyze the functionality of the neurons. We present here the first data of this innovative protocol, which can be considered a proof of concept. This type of modeling could offer many perspectives on understanding how these two large families of communicating cells interact. Our initiative will contribute to the study of the neural pathway of the microbiota-gut-brain axis and to efforts to test interventional molecules to modulate this physiological axis.

## 2. Results

### 2.1. Characterization of Human Glutamatergic Neurons and Dendritic Cells in a Microfluidic Device

The objective of the study is to propose an experimental model allowing (i) the association of human neuronal and immune cells in microfluidic devices and (ii) access to a non-invasive analysis method to study the communication between these two cell types in the context of inflammation to model neuro-immune interaction in the inflammatory condition of IBD.

Human cells were cultured in an asymmetric microfluidic device called DUALINK Shift in static conditions without pumping equipment (Figure 1A, Appendix A). The microfluidic device is made of Polydimethylsiloxane (PDMS), is biocompatible, and is optically transparent (for more details, see the material section). The device is composed of three compartments separated by microchannels. Each channel is fluidly isolated thanks to the architecture of the microchannels (3 µm high and 10 µm wide), allowing neurites and axons to project into them. Compartments 1 (channel 1) and 3 (channel 3) are where human glutamatergic neurons are cultured (Figure 1A, Appendix A). Compartment 2 (channel 2) is an asymmetric channel that isolates the soma and neurites using microchannels to create synaptic communication in vitro. Compartment 2 is closer to compartment 1, around 110 µm from channels 1 to 2 and about 510 µm from channels 2 to 3 (Appendix A).

To establish the cell culture protocol on-chip, only a culture of human glutamatergic neurons derived from innate pluripotent stem cells (iPSCs) was performed in the microfluidic device for 21 days before adding MoDCs (Figure 2A). To assess the culture in microfluidic devices, we confirm by immunofluorescence that glutamatergic neurons express MAP2, a marker of neuronal differentiation (Figure 1B), vGlut1, a presynaptic protein responsible for glutamate transport into synaptic vesicles (Figure 2C), and Beta III Tubulin, a principal constituent of microtubules (described in Appendix A). These cells showed a decrease in pluripotency markers over time and the expression of specific glutamatergic neuron markers in microfluidic devices (Appendix A), highlighting optimal maturation (previously published data, [16]). The MoDCs cells were prepared by isolating CD14+ cells from a healthy donor leuko-platelet concentrate. Before adding MoDCs on-chip, a Fluorescent Activated Cell Sorter (FACS) analysis was performed, showing an increase in CD209 and a decrease in CD14 expression (Appendix A) to confirm their phenotype. The Morphology of MoDCs is confirmed by the visualization of pseudopods in confocal microscopy (Appendix A). On day 21, human neuronal cells were exposed to mature dendritic cells (DCs) derived from monocytes (MoDCs). The presence of the MoDC in the microfluidic device is shown in transmitted light microscopy (Figure 1C). This observation is confirmed based on immunostaining 4 h after seeding these MoDCs in the microfluidic devices (Figure 1D). The CD209 phenotype is maintained as presented in Figure 1D (green in microfluidic channel 1 in this illustration), confirming that MoDCs conserved this phenotype 4 h after seeding with human glutamatergic neurons. The neuron-specific enolase (NSE) marker was assayed by Enzyme-Linked Immunosorbent Assay (ELISA) in the supernatant 4 h after co-culture, showing no neuronal cell damage (Figure 1E and Appendix A). The basic level of NSE in the culture is considered for the channel that neither receives LPS nor MoDC. Levels measured in channels exposed to lipopolysaccharide (LPS) or MoDC do not present an increased concentration of NSE, as shown in Figure 1E (and Appendix A). An ELISA assay performed after 4 h of co-culture reveals high IL-6 levels in the culture supernatant, suggesting a persistent ability of MoDCs to model an inflammatory condition (Figure 2B).

### 2.2. Neuro-Immuno Communication between Human Neurons and Immune Cells in Microfluidic Devices Is Measurable

The asymmetry of this microfluidic device is essential to isolating synapses and having a dendritic-axonal synapses compartment (Appendix A). Since the distance between channels 1 and 2 is around 110 µm, axons and dendrites can pass through, while the distance between channels 2 and 3 is about 510 µm, only axons can grow throughout the channel. Thus, to assess neuro-immune communication, different conditions have been used (Figure 2A). MoDCs were added to glutamatergic neurons on day 21, either in the cell body compartment (soma + axons) in channel 1 (condition 02) or channel 3 (condition 03). To assess whether the communication was taking place via axonal projections, MoDCs were added to channel 2, called the synaptic compartment (condition 05). As a control, LPS was added to channel 1 (condition 04) and channels 1 and 3 (condition 01).

To measure the effect of adding the MoDCs to the neuron culture, we chose to analyze the culture supernatant, the electric field recorded for 4 h after the co-culture, and the culture within the device by immunofluorescence staining.

The analysis of the culture supernatant after 4 h by ELISA reveals that MoDCs cells are secreting IL-6 when placed in the microfluidic device (Figure 2B). When neurons are exposed to LPS, no IL-6 is detected (Figure 2B). The IL-6 concentration in MoDC’s seeded channels is similar (channel 1: 1.3 × 10^3^ pg/mL, channel 2: 1.2 × 10^3^ pg/mL, and channel 3: 1.2 × 10^3^ pg/mL). These data highlight fluidic isolation: no passage of a compound from one microfluidic channel to the other by passive diffusion. The measure of the basal level in the opposite channel of all conditions, which only contains neurons, also confirms that neurons do not secrete IL-6 in response to adding MoDC. Similar concentrations between channels containing neurons cell bodies (channels 1 and 3) and channel 2 only containing neurons fibers also reveal that neurons do not secrete IL-6 in response to exposure to MoDC.

In parallel, the expression level of vGlut1 has been evaluated by immunostaining to assess the increasing glutamate secretion (Figure 2C,D). The analysis of vGlut1 immunostaining shows an increase in the marker percentage of fluorescence when immune cells are present. Concretely, adding MoDCs and LPS to glutamatergic neurons in channel 1 showed an equivalent percentage of vGlut1 expression independently of the stimulus (Figure 2C,D). Analysis of channel 2 shows an increase in the rate of vGlut1 expression when MoDCs are added but a decrease when LPS is added. In channel 3, an increase in the rate of vGlut1 expression was observed when LPS was added in channel 1, but not when MoDCs were added in channel 1 (Appendix A).

Conversely, adding MoDCs in channel 3 increased the percentage of vGlut1 expression in channel 2 but made no difference in the opposite channel (channel 1, Appendix A). These data differ from those obtained in electrophysiology (Figure 2E) and suggest electrical communication, not glutamate release, between the two opposite channels. According to further study conditions, these data could be confirmed with glutamate release analysis in the supernatant to have a kinetic response.

To evaluate the impact of the co-culture functionality of glutamatergic neurons and MoDCs, MEA was used. The device was coupled with electrodes to monitor cell-cell communication and record the electrophysiological activity of neurons. The activity is recorded and analyzed over a diagram representing the voltage variation over time (Appendix A). The functional activity could be expressed with a mean firing rate (MFR) corresponding to the number of spikes per second during the recording. Here, we show the percentage change in MFR before (baseline) and after the addition of MoDC to glutamatergic neurons (Figure 2E). Compared to baseline activity recorded on Day 20 (Appendix A), adding LPS increased glutamatergic activity in both channels with a mean of 25% to 125% of the increase (Figure 2E). In contrast, when MoDC were co-cultured in the same channel with glutamatergic neurons, their activity decreased (about −70%, in conditions 2 and 3). However, the functional activity of glutamatergic neurons in the opposite channel was increased (about 50% in conditions 2 and 3). These data are not statistically significant; however, there are tendencies either in the individual analysis of each experiment or in the average of the two replicates. This suggests a specific pattern of electrical activity in the presence of neuron-immune cell co-culture. These results indicate an increase in nerve impulses when MoDCs are present in channel 1 of the device. This difference allows us to conclude that electrical activity is involved in the communication between the two cell types and could be an excellent marker to assess the impact of immunological signaling pathways on neuronal activity.

## 3. Discussion

The physiopathological mechanisms of IBDs are possibly linked to the microbiota-gut-brain axis that should be impaired in these disorders. It is necessary to understand this bidirectional communication to propose specific innovative therapies. Most studies focus on the brain to look for evidence that explains behavioral manifestations [17,18]. Hypotheses about inflammation consequences in IBD patients’ brains are about a damaged blood-brain barrier and damage to endothelial cells’ tight junctions, allowing endotoxins and inflammatory mediators from the gut to reach brain tissue [19]. Inflammatory activity in the cerebral parenchyma results in a decrease in neurogenesis and the activation of neurodegeneration. Mechanisms involving microglial cells’ activation, cytokine neurotoxicity, and reactive oxygen species are mainly localized in the hippocampus in IBD patients [20]. Furthermore, some particularities in the brain structures of IBD patients have been shown by structural and functional magnetic resonance imaging in the limbic system, basal ganglia, and hypothalamus [18,21]. IBD research has shown that inflammation can have local effects on the ENS and sensory neurons within the intestinal tissue [22]. Our study aims to explore how inflammatory signals from immune cells can be transmitted to nerves. The neuroimmune cell unit plays an essential role in regulating the homeostasis and physiology of tissues, particularly the intestine [23]. Our team has taken the initiative to model this unit in a microfluidic device using electrophysiology to measure the effect of immune signals on neurons. While many existing gut-on-chip models exist, none include nerves [14]. To the best of our knowledge, our project is the first to demonstrate the feasibility of a microfluidic chip that incorporates the cells involved in the neuroimmune cell unit of the gut.

This preliminary study aims to elaborate a microfluidic platform to understand communication mechanisms between nerve and immune cells using a tool to study cell-cell interaction. We develop a first platform involving MoDC chosen to represent a robust model of innate immune cells capable of transmitting an inflammatory signal to other cells (T and B cells). To reproduce inflammatory conditions found in the mucosa of IBD patients exposed to pathogens, MoDCs were exposed to LPS (1 µg/mL) from *E. coli* 12 h before seeding. The immune cells were previously characterized (Appendix A) and showed a dendritic cell phenotype with CD209 expression [24]. Glutamatergic neurons were chosen for their high electrophysiological activity, making these cells a model of choice. Glutamatergic neurons were maintained under these conditions (after two recordings and in co-culture conditions with MoDCs) and expressed the classically expected phenotypic markers already published by our team [16]. These cells express vGlut1, a presynaptic protein responsible for glutamate transport into synaptic vesicles. They are found in functional glutamatergic neurons and Beta III Tubulin, a major constituent of microtubules, and play a critical role in axon guidance and maintenance (Figure 2C,D and Appendix A). We showed that neuronal functional activity could be a marker to characterize communication between immune cells primed by bacterial compounds and neurons. Adding primed immune cells to a functional neuronal culture triggers neuronal communication between neurons by electrophysiology. This device offers the possibility to study various microbial molecules, peptides, probiotics, and therapeutics for assessing their effect on mucosal immune cells and their secondary effect on nerves. Gnotobiotic mice are currently the standard model to study the impact of microbiota on in vivo physiology [25]. In vivo studies are time-consuming, expensive, and involve ethical considerations due to the high level of complexity that could interfere with understanding subtle mechanisms. Some initiatives also consist of studying ex vivo explants from the colon or the gut to study interactions with the microbiota and the physiology of the axis [26]. The biological phenomenon’s complexity is approached by the initiative of gut-on-chip modeling of a lumen and a whole mucosa with villi [27]. Complex initiatives to model the gut-brain axis involve multi-chambers with different Organs-on-Chip and barrier interfaces to reach the axis’s complexity level in in vitro conditions [28]. Microfluidic technology allows the modeling of tissue with tight control of fluidic conditions and avoids the complexity of reaching samples from the microenvironment for assays [29]. Our microfluidic device represents a tool to study cell-cell communication through soluble factor assays and electrophysiological activity, bridging the gap between in vitro and in vivo and proposing a new way to study gut-brain communication thanks to neuron electrophysiology. Conventional cell culture techniques struggle to reproduce paracrine communication between two cell types. We propose an in vitro model for nerve interaction with peripheral immune cells, which differs from many previous microfluidic Organ-on-Chip models that focus on modeling brain structures for studying the effects of neuroinflammation [30]. We also analyze the fluidic isolation of culture channels in the absence of IL-6 passage from one compartment to another. Microfluidic compartmentalization allows finely controlled cell microenvironments, chemical stimuli, and communication monitoring. Furthermore, the total volume of the channel is around 100 µL, so even small amounts of secreted factors are not fully diluted and, therefore, detectable. Furthermore, microfluidic systems allow the control of fluids due to laminar fluid flow. The focus on electrophysiological response to stimuli allows us to measure a rapid answer, the strength of the signal by interpretation of the firing rate, and the amplitude of the action potentials. We evaluated the possibility of studying functional responses in one channel after adding cells in the opposite channel by electrophysiology. An increase in IL-6 in the MoDC-containing compartment correlated with a decrease in electrical activity, with an inverse increase in the electrical activity in the opposing channel. MEA technology allows extracellular recording, unlike the patch clamp, which records intracellular activity. It also represents the optimal approach to studying the functional activity of neurons in networks [31]. This study showed that obtaining compartmentalized cell cultures in a microfluidic device was possible. MoDCs were chosen for their ease of differentiation from human peripheral blood mononuclear cells and their ability to respond to LPS signals by TLR2. DCs are crucial immune cells for immunity and tolerance and are involved in developing the pro-inflammatory state of the mucosa in IBD by inducing Th1 and retinoic acid metabolism [32]. The intestinal microenvironment regulates DCs through the gut microbiome and the cellular environment of the mucosa [16]. DCs were shown to interact with sensory neurons in vitro through the intermediary of tunneling nanotubes [33].

Future studies of our devices will focus on the biological mechanisms involved in neuro-immune interactions. The following steps will be taken: the complexification of the in vitro model to reach a gut-brain axis on-chip; the integration of epithelial cells to achieve a better model of a complete intestinal mucosa; and the proposal to carry out studies on bacterial components or molecules for therapeutic purposes. To generalize our results, additional innate immune cell types already known to be related to nerves in the gut mucosa must be studied in our chip. As we know, macrophages are closely localized to nerve fibers and are involved in homeostasis and pathology [34]. In inflammatory bowel syndrome, there is evidence of dysregulation of interaction loops between mast cells and enteric neurons within the mucosa. Furthermore, innate lymphoid cells are closely related to the enteric nervous system and glial cells and are implicated in the mucosal regulation of pathogens and inflammation [35,36].

This study needs to be reproduced with a higher number of repeats to reach a statistical significance rate representing this exploratory study’s principal limit and to understand the triggering of electrophysiologic signals in neurons by exposing neurons only to IL-6.

Our study highlights the existence of a functional neuro-immune cell unit through in vitro experimentation. We found that immune cells transmit an inflammatory signal to neurons, triggering action potentials in response to MoDC addition. Our device models a segment of the microbiota-gut-brain axis as it isolates the communication between nerves and immune cells of the intestinal mucosa, providing a tool to investigate the biological mechanism of this interaction. Our focus on IBD has led us to explore additional methods for approaching the gut-brain axis in pathology, as described in the last review by Moysidou and Owens [37]. This new approach complements our existing methodologies and allows us to better understand the complex relationship between immunity and neurons. Our concept also enables the study of inflammation from the perspective of electrophysiology. Microfluidic devices are one tool that can precisely study the highly complex biological mechanisms involved in IBD between immune cells and neurons. Our device’s development and functional study represent a dynamic effort to reduce the use of animals, optimize cost and efficiency, and improve tools for functional biology research [38].

In conclusion, this proof-of-concept study has shown its ability to respond to our requirements with a functional neuronal culture, a device to receive living immune cells, and a multimodal way to assess immune and nervous communication: electrophysiology, supernatant assay, and vesicle labeling. Our results have shown, for the first time, that the transduction of an inflammatory immune signal to neurons leads to an electrical impulse materialized by an increased firing rate in neuronal culture. This signal is bottom-up, consistent with the intuitive neuroimmune communication within the intestinal mucosa. Literature reports evidence about a neuron’s paracrine response to regulate immune cells by neurotransmitters and peptides, representing the next step to be studied in this device [39]. Moreover, the device allows the study of the colocalization of immune cells and neurons and potential studies of direct interaction, possibly assimilating to a neuro-immune synapse. Developing organ-on-chip models for studying the gut-brain axis solves some challenges in in vitro and in vivo models.

## 4. Materials and Methods

### 4.1. Device Fabrication

The asymmetric microfluidic devices, called Dualink Shift MEA (N3XU_ASYM-FL), were fabricated according to NETRI’s standard protocol [15,16]. Briefly, SU-8 molds were fabricated using conventional photolithography techniques using the SU-8 photoresist series. Silicon wafers were diced and integrated into 3D-printed molds to create master molds. Master molds were replicated using dedicated silicone and polyurethane positive replicates. Polydimethylsiloxane (PDMS, Sylgard 184, Dow Corning, Midland, MI, USA) was prepared and cast onto the replicated molds before being cured at 65 °C for 2 h. The PDMS was then cut to the desired size before being peeled off the mold. The mixture is centrifuged, degassed, and poured into the device mold. The PDMS is then cross-linked at 80 °C for 1 h, then cut and removed from the mold. Then the microfluidic chip is covalently bonded to a polystyrene substrate aligned in SBS format (Society for Biomolecular Screening format), previously rinsed with isopropanol, and activated with O_2_ plasma. Finally, the devices are placed at 100 °C for 10 min and filled with 70% ethanol to be sterilized.

### 4.2. Culture and Maintenance of Human iPS-Derived Glutamatergic Neurons

The platform allows the growth of several cell types in each microfluidic channel under static conditions and without pump equipment while monitoring the impact of human immune cells on glutamatergic neurons via electrophysiological activity and secreted factors. On this platform, we cultured glutamatergic neurons derived from induced pluripotent stem cells (IPSCs). Glutamatergic neurons were seeded in channels 1 and 3 to reach 1000 cells/mm^2^. Human-derived materials were preserved and handled with the approval and under the guidelines of French legislation. The accreditation number related to the use of human materials is DC-2020-4203. Human-induced pluripotent stem cell-derived cortical glutamatergic neurons (BrainXell, BX-0300, Madisson, WI, USA) are maintained in channels 1 and 3 compartments for up to 21 days. Microfluidic devices were coated with PDL (0.05 mg/mL) overnight at 37 °C under 5% CO_2_. Glutamatergic neurons were seeded by placing 10 µL of a 1 × 10^7^ cells/mL neuron suspension in the inlet reservoir of channels 1 and 3 in an asymmetric device for a targeting density of 600 cells/mm^2^. Cell culture media were replaced every 2 to 3 days with provider media, as previously described [16]. Neurons were cultured for up to 21 days under a controlled environment (37 °C and 5% CO_2_).

### 4.3. Culture and Preparation of Human MoDCs

Human MoDCs were used for this experiment. Peripheral blood mononuclear cells (PBMCs) were obtained from the buffy coats of healthy donors (EFS Auvergne-Rhône-Alpes) by density gradient centrifugation over a lymphocyte separation medium (Eurobio; Abcys, Courtaboeuf, France). Monocytes (CD14+) were positively sorted from PBMC with magnetic beads (Miltenyi Biotec, Paris, France). The culture of monocytes induced differentiation of MoDC with complete RPMI medium (RPMI 1640, 10% FBS, 1% penicillin/streptomycin) supplemented with 100 ng/mL of huGM-CSF (Miltenyi Biotec, Paris, France) and 50 ng/mL of huIL-4 (Miltenyi Biotec, Paris, France) for six days, with the addition of fresh medium on day 4. The monocyte differentiation into MoDC was checked by microscopy, which visualized the formation of villosity. MoDC were cultured at 1 × 10^6^ cells/mL in 12-well plates in a complete RPMI medium associated with treatment with 50 μg/mL of polymyxin B (InvivoGen, San Diego, CA, USA). A total of 1 µg/mL of lipopolysaccharide (LPS) from *E. coli* (Sigma-Aldrich, St Louis, MO, USA) was added to prime immune cells. It induced their activation for 4 h before seeding in the microfluidic device.

### 4.4. LPS and MoDC Additions

Culture media was removed for conditions 01 and 04, and glutamatergic neurons were treated with LPS from *E. Coli* (2 µg/mL, Sigma-Aldrich, St Louis, MO, USA) diluted in neuron culture media. Then, cells are incubated for 4 h under a controlled environment (37 °C, 5% CO_2_). Cells were washed with neuron culture media before recording.

For conditions 02, 03, and 05, MoDCs were washed from their culture media and LPS by two centrifugations (1500× *g* rpm, 4 °C, 10 min). Then MoDCs were suspended in neuron media culture at a seeding concentration of 10 M cells/mL. MoDCs were incubated with glutamatergic neurons and maintained for 2 days under a controlled environment (37 °C, 5% CO_2_). The first recording was performed 4 h after MoDC seeding.

### 4.5. Immunoassays

The supernatant was collected 4 h after LPS or MoDC addition to the glutamatergic neurons in each channel by pipetting into the fluidic outlet and freezing at −80 °C until use. Two factors released have been analyzed in the cell culture supernatant by ELISA thanks to the commercial kit: IL-6 (MBS175877-96, MyBioSource, San Diego, CA, USA) and NSE (MBS161973-48, MyBioSource, San Diego, CA, USA). Samples were diluted ½ for the IL-6 assay using the sample diluent of a commercial kit. No dilution was performed on samples for the NSE assay. Data analysis has been done with GraphPad Prism.

### 4.6. Electrophysiological Recording

The electrophysiological recordings have been performed thanks to a commercial system (MultiChannel Systems, Reutlingen, Germany). A first recording of 10 min on glutamatergic neurons was done on day 20. Depending on conditions, a second recording was performed after 12 h of MoDC neuron co-culture or upon LPS (4 h) addition. All data were recorded with a MEA2100-256 system, commercially available from MultiChannel Systems (Reutlingen, Germany), composed of a 256-channel amplifier head-stage. The recording was performed using commercially available software (Multi Channel Experimenter, MultiChannel Systems). All the experiments were carried out with 256MEA100/30iRITO-w/o (MCS) that consist of 30 μm diameter electrodes spaced by 100 μm. Raw data were offline filtered by a Bandpass Butterworth filter of second order (with cut-off frequencies of 100 Hz and 2500 Hz). Electrodes that were outside of cell culture channels have been excluded from analyses.

### 4.7. Electrophysiological Data Processing

Signal records have been analyzed with MultiChannel proprietary and open-access software (Multi-Channel Analyzer, MultiChannel Systems, Reutlingen, Germany). Spike detection was performed with a threshold of 5× the standard deviation of the signal on noise. The mean firing rate (MFR) is the number of spikes per second from each channel and was calculated based on the active electrode signal divided by recording time (10 min). An active electrode has been considered when the electrode detects at least 0.1 spikes/s. Data analysis has been done with GraphPad Prism. The electrophysiological studies included the compartmentalized part of the chip, excluding activity localized under the microchannels to avoid an overestimation bias in the results. Indeed, the microchannels contain the neurons’ axons and concentrate the neurons’ functional activity. This activity is isolated and amplified at this location.

### 4.8. Immunostaining Protocol

Cultures were fixed in 4% paraformaldehyde (PFA) for 30 min at room temperature, as previously described [16]. Cells were washed three times with PBS and permeabilized for 10 min with 0.1% Triton-X100, followed by 30 min with 3% BSA. Primary antibodies were added, and the devices were incubated overnight at 4 °C. The cells were rinsed three times with PBS and further incubated with the corresponding secondary antibodies for 2 h at room temperature (references and working solutions of antibodies used in the study are listed in Table 1). For the quantification of vGlut1, the percentage of the stained surface was quantified on a defined area of the device for all three channels using Image J. For the quantification of Sox2 and Nestin, the number of total DAPI-labeled cells was quantified and related to the number of positively labeled Sox2 or Nestin cells. Data analysis has been done with GraphPad Prism (GraphPad, San Diego, CA, USA).

Images were acquired with an inverted epifluorescence microscope, the AxioObserver 7 (Zeiss), fitted with a CMOS camera.

## Figures and Tables

**Figure 1 ijms-24-10568-f001:**
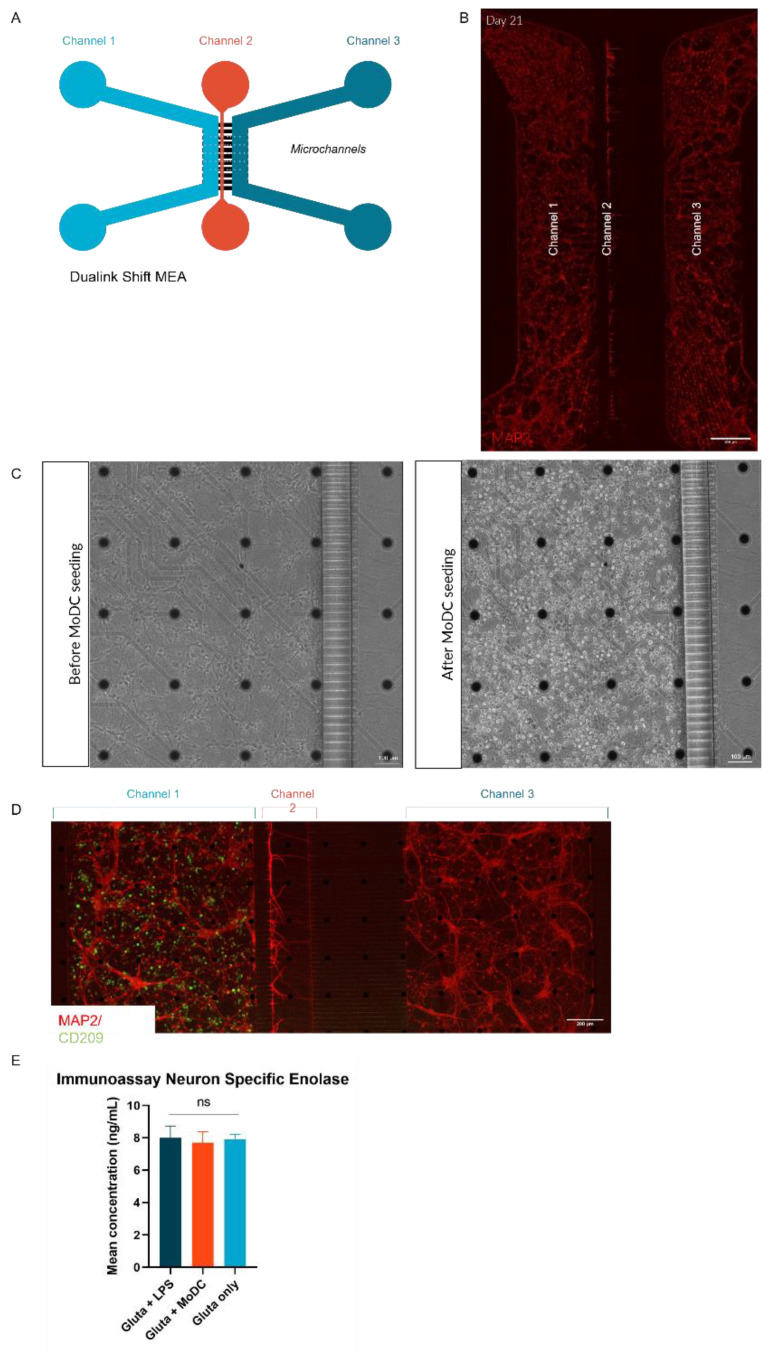
Characterization of human immune and neuronal cell co-culture in an asymmetric microfluidic device. (**A**) Schema of the asymmetric microfluidic device, called Dualink Shift MEA, composed of three compartments. (**B**) Human cortical glutamatergic neurons in a microfluidic device stained red with an anti-MAP2 antibody. (**C**) Visualization of the cells before and after adding MoDCs to the microfluidic device in transmitted light microscopy. (**D**) Immunofluorescence staining of MoDC cells colored green with antibody anti-CD209 in channel 1 and glutamatergic neurons with antibody anti-MAP2 seeded in channels 1 and 3. (**E**) Graph of the quantification of NSE in the pooled supernatant of microfluidic devices as a function of conditions. FACS: fluorescent activated cell sorter; MEA: microelectrode array; MoDCs: monocyte-derived dendritic cells; NSE: neuron-specific enolase.

**Figure 2 ijms-24-10568-f002:**
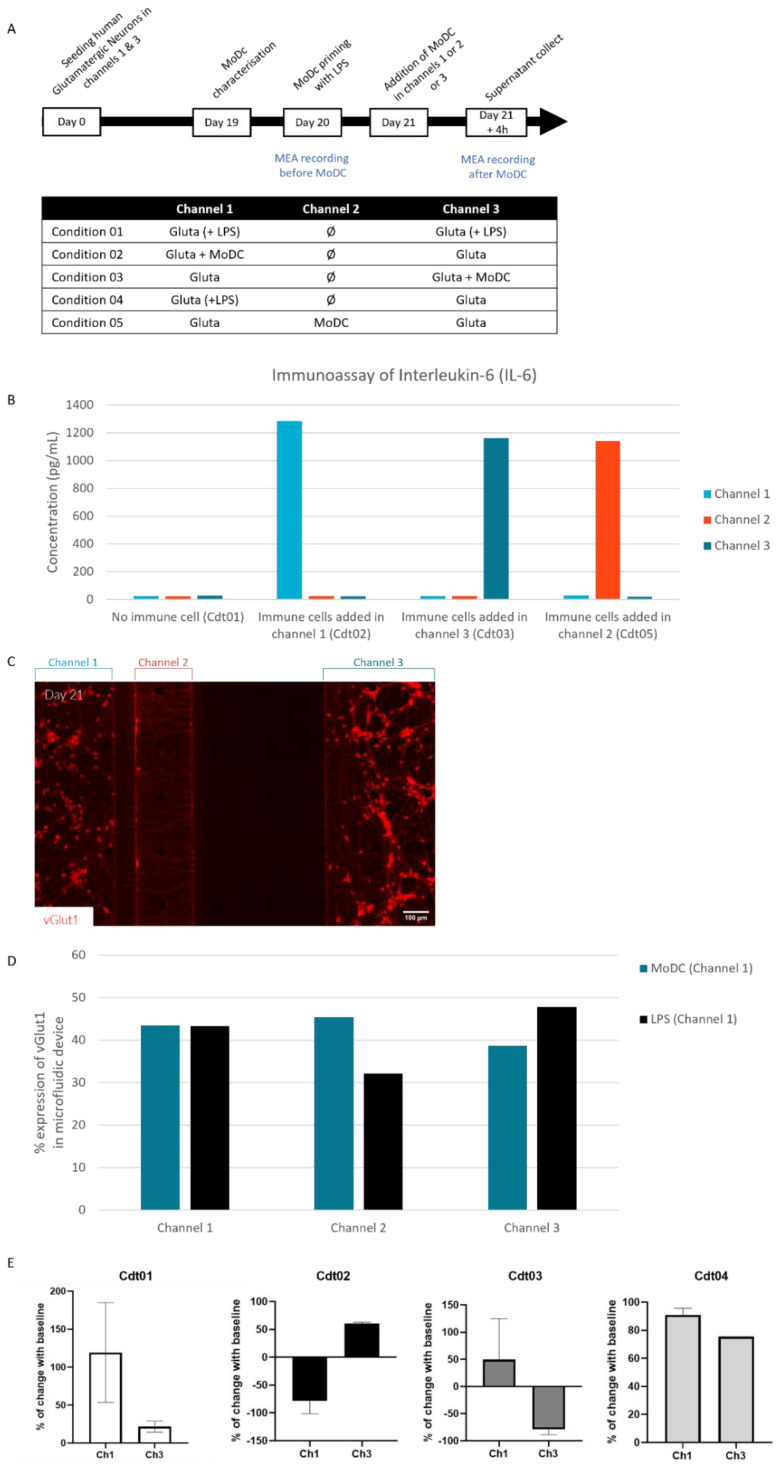
Characterization of the neuro-immunological communication in the microfluidic device. (**A**) Experimental design: seeding glutamatergic neurons at day 0 in channels 1 and 3 of the microfluidic devices, adding MoDC cells at day 21, and recording electrical activity via MEAs with detailed descriptions of the conditions. (**B**) Graph of ELISA immunoassay analysis of interleukin-6 in the supernatant of each channel (pooled from chips) in the presence of MoDCs or LPS or with glutamatergic neurons only. (**C**) Immunofluorescent pictures of glutamatergic neurons at day 21 stained with anti-vGlut1 (Red). (**D**) Semi-quantification of vGlut1 immunostaining according to the conditions. (**E**) Graphs represent the percentage of electrophysiological activity change from baseline (before adding MoDC) to after adding MoDC in channels 1 and 3 according to the different conditions in the MEA microfluidic device. ELISA: enzyme-linked immunosorbent assay; LPS: lipopolysaccharide; MEA: microelectrode array; MoDCs: monocyte-derived dendritic cells.

**Table 1 ijms-24-10568-t001:** List of immunofluorescent antibodies.

Antibodies Name	Supplier/Reference	Final Concentration
Nestin	ThermoFisher/14-9843-82	1 µg/mL
Sox2	Merck/AB5603	2 µg/mL
vGlut1	Synaptic Systems/135304	5 µg/mL
MAP2	Abcam/ab96378	1 µg/mL
Beta-III-Tubulin	ThermoFisher/MA1-118	5 µg/mL
DC-SIGN/CD209	Bio-techne/DDX0207-100	0.05 µg/mL

## Data Availability

Data are available on request to the corresponding authors.

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
