# Peer review of "Proof-of-Concept Human Organ-on-Chip Study: First Step of Platform to Assess Neuro-Immunological Communication Involved in Inflammatory Bowel Diseases"

_ijms, 2023, doi:10.3390/ijms241310568_

Round 1
Reviewer 1 Report (Previous Reviewer 3)
The authors have diligently revised their manuscript, taking into account the valuable feedback provided. They have meticulously incorporated the suggested revisions and enhancements to ensure the manuscript meets the highest scientific standards. This process has allowed the authors to refine their work and address any potential gaps or areas of improvement, ultimately strengthening the overall quality of the manuscript.
Author Response
Dear Editor and Reviewers,
Thank you for taking the time to review our work. Please find in the following letter the details of our modifications according to the reviewer’s comments. We submit you a marked version and an unmarked version of our manuscript, figures, and supplementary figures that are also available attached.
We greatly appreciate your work to improve our manuscript. We sincerely hope this revised version is more precise and will meet your expectations.
Please find below a point-by-point response to each of the referee’s comments and a description of the changes made.
Reviewer 2:
- Add more data as the paper is looking short:
Our manuscript proposal is focused on a short communication format. While our data is preliminary, we have provided a thorough description. To provide more detail on additional figures, we suggest including expanded explanations in the body of the text. Specifically, we recommend highlighting explanations in gray for lines 113 to 124 and highlighting in pink for lines 127-131 for Figure S1, adding more detail from lines 127 for Figure S2 and from lines 215 to 217 for supplemental Figure S7. The descriptions for the other additional figures have already been provided in the text. Following your recommendations, we have added more details about the microfluidic device and more about the bibliographic context of our study.
- Revise the bar graphs in origin pro or some other software with at least 300 dpi:
Thank you for your comment. The diagram has been revised and performed using GraphPad Prism (addition in Material and Method section).
- Clearly show the novelty of the work:
We have made some updates to the discussion section and have added new bibliographical references. These changes can be found on lines 260-269 and 306 to 309, highlighted in yellow. We hope these updates will address any concerns you may have had.
- Show a comparison with the existing literature research gap and this work worth in terms of addition or improvement to works were already done:
In the same way as the previous question, we have added a few references to the introduction section from 74 to 89, highlighted in red, reporting the actual missing in vitro IBD models.
- Organ-on-a-chip concepts are mainly based on the microfluidics approach, but there is no detail in this paper:
Thank you for your comment. The characteristics of the microfluidic device are shown in Figure 1 and Figure S1 (supplementary data). We have added more precision in the body of the text from lines 113 to 124, highlighted in gray.
- Mixing, separation, and cell analysis which one is used in this work :
Thank you for your comment. Indeed, the protocol wasn't detailed enough in the body of the text, particularly regarding the timings for adding cell types during culture. We have described this in the body of the text from lines 125 to 127 and 134-136, highlighted in green, by adding time qualifiers for each cell type. In the body of the text, we have also referred to Figure 2A, which schematically describes the experimental protocol (line 127).
- What are the practical or real time application of this concept:
Thanks for your comment; we added a paragraph about perspectives and application of the presented work from line 347 to line 361, highlighted in blue, with new bibliographic references.
- Can you add some more data and make it a little bit more realistic:
As presented in answer to the first remarks, all our data are presented here. We hope that the new parts added in blue/green/ and yellow, and purple will go in the same direction. To increase the consistency of the results, we exposed the motivation of our work, as asked in previous remarks, and on the recorded data exploited in the text highlighted in purple.
- Figure one shows experimental work where is a detail of fabrication, and material preparation:
Thank you for your comment. The characteristics of the microfluidic device are shown in Figure 1, Figure S1 (supplementary data), and in the material and method section and from lines 113 to 124, highlighted in gray (cf. answer to question #5).
- Can you add some simulation work to this paper?:
While certainly interesting, the authors respectfully remind the reviewer that this kind of work is out of the scope of this paper. Moreover, considering the type of flow present in these devices (ultra-laminar and gravity-driven), the authors do not think such work would bring any insights.
Sincerely,

Reviewer 2 Report (New Reviewer)
This paper presents
Inflammatory bowel diseases (IBD) are complex chronic inflammatory disorders of the gastrointestinal (GI) tract. Recent evidence suggests that the gut-brain axis may play a pivotal role in gastrointestinal and neurological diseases, especially in IBD. Here, we present first proof of concept for a microfluidic technology to model this bilateral neuro-immunological communication. We designed a device composed of three compartments with an asymmetric channel that allows the isolation of soma and neurites thanks to microchannels and creates an in vitro synaptic compartment. Human-induced pluripotent stem cell-derived cortical glutamatergic neurons were maintained in soma compartments for up to 21 days. We performed a localized addition of dendritic cells (MoDCs) on either soma or synaptic compartment. The microfluidic device was coupled with MicroElectrode Arrays (MEAs) to assess the impact on the electrophysiological activity of neurons during the addition of dendritic cells. Our data highlight that an electrophysiologic signal was transmitted between two compartments of glutamatergic neurons linked by synapses in a bottom-up way when soma is exposed to primed dendritic cells. In conclusion, our study authenticates a communication between dendritic cells and sensitive neurons transmissible to the nervous system in inflammatory conditions, as in IBD. This platform opens the way to complexification with gut component, to reach a for pharmacological compound screening in blocking the gut-brain axis at a mucosal level and may help patients.
1. Add more data as the paper is looking short
2. Revise the bar graphs in origin pro or some other software with at least 300 dpi
3. Clearly show the novelty of work
4. Show a comparison with existing literature research gap and this work worth in terms of addition or improvement to the works already done
5. Organ on a chip concepts are mainly based on the microfluidics approach but there no detail in this paper
6. Mixing, separation and cell analysis which one is used in this work
7. What are the practical or real-time applications of this concept?
8. Can you add some more data and make it a little bit more realistic?
9. Figure one is showing experimental work where is a detail of fabrication, and material preparation.
10. Can you add some simulation work to this paper?
Author Response
Dear Editor and Reviewers,
Thank you for taking the time to review our work. Please find in the following letter the details of our modifications according to the reviewer’s comments. We submit you a marked version and an unmarked version of our manuscript, figures, and supplementary figures that are also available attached.
We greatly appreciate your work to improve our manuscript. We sincerely hope this revised version is more precise and will meet your expectations.
Please find below a point-by-point response to each of the referee’s comments and a description of the changes made.
Reviewer 2:
- Add more data as the paper is looking short:
Our manuscript proposal is focused on a short communication format. While our data is preliminary, we have provided a thorough description. To provide more detail on additional figures, we suggest including expanded explanations in the body of the text. Specifically, we recommend highlighting explanations in gray for lines 113 to 124 and highlighting in pink for lines 127-131 for Figure S1, adding more detail from lines 127 for Figure S2 and from lines 215 to 217 for supplemental Figure S7. The descriptions for the other additional figures have already been provided in the text. Following your recommendations, we have added more details about the microfluidic device and more about the bibliographic context of our study.
- Revise the bar graphs in origin pro or some other software with at least 300 dpi:
Thank you for your comment. The diagram has been revised and performed using GraphPad Prism (addition in Material and Method section).
- Clearly show the novelty of the work:
We have made some updates to the discussion section and have added new bibliographical references. These changes can be found on lines 260-269 and 306 to 309, highlighted in yellow. We hope these updates will address any concerns you may have had.
- Show a comparison with the existing literature research gap and this work worth in terms of addition or improvement to works were already done:
In the same way as the previous question, we have added a few references to the introduction section from 74 to 89, highlighted in red, reporting the actual missing in vitro IBD models.
- Organ-on-a-chip concepts are mainly based on the microfluidics approach, but there is no detail in this paper:
Thank you for your comment. The characteristics of the microfluidic device are shown in Figure 1 and Figure S1 (supplementary data). We have added more precision in the body of the text from lines 113 to 124, highlighted in gray.
- Mixing, separation, and cell analysis which one is used in this work :
Thank you for your comment. Indeed, the protocol wasn't detailed enough in the body of the text, particularly regarding the timings for adding cell types during culture. We have described this in the body of the text from lines 125 to 127 and 134-136, highlighted in green, by adding time qualifiers for each cell type. In the body of the text, we have also referred to Figure 2A, which schematically describes the experimental protocol (line 127).
- What are the practical or real time application of this concept:
Thanks for your comment; we added a paragraph about perspectives and application of the presented work from line 347 to line 361, highlighted in blue, with new bibliographic references.
- Can you add some more data and make it a little bit more realistic:
As presented in answer to the first remarks, all our data are presented here. We hope that the new parts added in blue/green/ and yellow, and purple will go in the same direction. To increase the consistency of the results, we exposed the motivation of our work, as asked in previous remarks, and on the recorded data exploited in the text highlighted in purple.
- Figure one shows experimental work where is a detail of fabrication, and material preparation:
Thank you for your comment. The characteristics of the microfluidic device are shown in Figure 1, Figure S1 (supplementary data), and in the material and method section and from lines 113 to 124, highlighted in gray (cf. answer to question #5).
- Can you add some simulation work to this paper?:
While certainly interesting, the authors respectfully remind the reviewer that this kind of work is out of the scope of this paper. Moreover, considering the type of flow present in these devices (ultra-laminar and gravity-driven), the authors do not think such work would bring any insights.
Sincerely,

Round 2
Reviewer 2 Report (New Reviewer)
The supplementary file is not arranged correctly. Please revise all slides to ensure clarity and full slides with clear descriptions for readers. Additionally, to improve the quality of the paper, consider adding the following references in the introduction section where microfluidics devices are mentioned: "Mixing, separation, and cell sorting are some processes commonly used in the organ-on-a-chip concept." It is also suggested to include a description of these processes in the introduction part, as it is currently missing. Lastly, consider incorporating the following reference: "Numerical analysis of non-aligned inputs M-type micromixers with different shaped obstacles for biomedical applications."
Author Response
Manuscript ID: ijms-2277517
International Journal of Molecular Science
Saint-Etienne, June 21 2023
Dear Editor and Reviewers,
Thank you for taking the time to review our work. Please find in the following letter the details of our modifications according to the reviewer’s comments. We submit you a marked version and an unmarked version of our manuscript, figures, and supplementary figures that are also available attached.
We greatly appreciate your work to improve our manuscript. We sincerely hope this revised version is more precise and will meet your expectations.
Please find below a point-by-point response to each of the referee’s comments and a description of the changes made.
In general, we have added more details concerning the manufacture of the device. We have added a more detailed description of the microfluidic device (grey part) to explain this.
- The supplementary file is not arranged correctly. Please revise all slides to ensure clarity and full slides with clear descriptions for readers.
Thank you for your comment. We have gone into more detail about the legends to help readers understand them better. We have also removed the titles of the figures in the "supplementary material" section of the manuscript.
- Consider adding the following references in the introduction section where microfluidic devices are mentioned: “Mixing, separation and cell sorting are some processes commonly used in the organ-on-a-chip concept”.
Thanks for your comment, but there is no “mixing, separation, and cell sorting” in this article. With all our respect, we apologize for not understanding your comment.
- Incorporate the following reference: “numerical analysis of non-aligned inputs M-type micromixers with different shaped obstacles for biomedical applications”.
The authors respectfully remind the reviewer that this type of work is outside the scope of this article. No claim about micromixers or mixing has been made in this study.
Sincerely,
This manuscript is a resubmission of an earlier submission. The following is a list of the peer review reports and author responses from that submission.
Round 1
Reviewer 1 Report
The paper deal with the development of a microfluidic device for modelling the neuro-immunological communication involved in IBS.
Even if the aim is timely, many concerns prevent the Reviewer to give a positive opinion, supporting the need of major modification to the text, that follows:
- TITLE: The Title is impressive, however is quite far from the content of the manuscript. The microfluidic based model proposed hosts only TWO cell lines, so it is very far from a human chip model, even if the cells are derived from human donors (monocytes) and iPS (glutaneurons). Furthermore the device allows investigating only the interaction between the two cell lines indicated, that is very far from the gut-brain based neuro-immunological communication that involves many different cell lines. I suggest to re-write the title to be more coherent with the experimental data.
- INTRODUCTION: The Introduction is very long and goes trough the inflammatory bowel disease, the gut-brain axis, the neuro-immune communication for about 85 lines, then only 10 lines are dedicated to the microfluidic device based models (that are quickly re-described in Discussion) and 5 lines to the aim of the study. I think that this paragraph should be re-written considering that the main aim is a microfluidic platform that wish to represent a useful new in vitro model.
- AIM: At the end of the Introduction, the Authors state that their study “aimed at developing andì in vitro platform to evaluate neuro-immune cell communication in compartmentalised microfluidic device”, and that “their first set of data could offer many perspectives to understand the MGB axis nervous route”. I think both the statements are too ambitious considering the content of the paper because: 1) with only TWO cell lines the model is very far from representing the complexity of the in vivo situation: many attempt has been done in this direction but state-of-the-art in vitro solutions are based on (3D) co-colture, multi-organ-on-a chip based system, with the aim of representing at least the key cell components/mechanical cues of the target organs (for example, for the BBB at least endothelial cells and astrocytes; for the brain neurons, astrocytes and microglia); 2) given that, it is difficult to accept that the obtained data can really help in evaluating the neuro-immune cell communication, but I suggest to highlight which biochemical route thay can really help to investigate. To be realistic, the proposed model is a microfluidic device that allow investigating some selected biochemical routes related to glutaneurons-dendritic cells.
- RESULTS: The Authors often use the word “validated” that I suggest to replace with “assess” (in all the text) because there are no referring standard to confirm that the cells/the device are performing as expected in the situation they model (i.e.: even if the cells express the expected markers, there are no indication that they act as expected respect to a standard/commonly used in vitro model representing the targeted in vivo situation). The Figure 1 contains a great amount of pictures, images, graph that are very difficult to read, and prevent the reader to evaluate the obtained data. In the Par 2.2. the Authors describe different testing situation but it is hard to understand: Why they select that experimental set-up? What they want to assess/prove by comparing all the described set-up?. Finally it is not clear when/if the cells are in dynamic or in static conditions within the microfluidic device in all the described experimental conditions (media flow rate?).
- DISCUSSION: In the Discussion the minus/limits of the device and of the present study are underestimated/not really critically addressed: for example, the need to include in the device more than two cell types is never critically discussed and the state-of-the art/potential DIRECT competitor are not considered/described (only a general discussion about microfluidic technology/conventional cell cultures is included). In the last sentences, (from “In conclusion”, line 27) they summarise the relevance of the results in a more realistic and interesting way, but again a detailed critical description is missing.
I strongly suggest to re-consider the focus of the paper in order to present the device and the data in a more realistic way to really give the opportunity to the readers to understand the plus/minus/limits of the proposed device (and biological models selected) and its potential to really contribute to the field of in vitro innovative models for MGA routes investigation.
Reviewer 2 Report
Comments:
In this article, Gabriel-Segard et al aimed to study the gut-brain neuro-immunological communication involved in IBD with a microfluidic device they developed. However, the data presented is very unclear and vague to support their hypothesis. The major problem is their design of experiments that could not model the IBD conditions as they claimed. Their protocols did not include any components of the gut and the co-culture only involved the brain cortical neurons and dendritic cells (MoDC). Their device may allow the study of neuro-immunological interaction within the brain but not the more complex gut-brain communication. Also, the descriptions in the Results section were different from what they have shown in the figures and they missed a lot of important experimental details. This makes the audience hardly understand exactly what message they would like to deliver in their results. Most of the experiments were not done in replicates and no significant conclusion can be drawn from their findings. Their data were very preliminary and not suitable for publishing in the present status. In addition, the Supplementary were not provided.
Major concerns:
1. The main text of the mentioned about immunofluorescence (IF) of vGlut1 and Beta III Tubulin in Figure 1B., but the figure legend said it was IF of MAP2. So, what was shown in Figure 1B? vGlut1? Beta III Tubulin? MAP2? Also, for Figure 1F, they mentioned about IF in the main text but they only showed phase-contrast image in the figure.
2. No description of Figure 1C, 1D, IE in the Results section. Please describe every figure shown and interrupt them in the result.
3. In Condition 02 and 04, the neurons were co-cultured with MoDC. What medium was used for maintain the culture for 2 days? The culture medium used for maintain neurons is different from the medium for maintaining MoDC culture. The authors only measured NSE level to show there was no increased neuronal cell death. How about MoDC? Did they survive during the co-culture? It is important to provide the experimental details as the authors claimed it is the very first protocol for such a study.
4. How was the IL-6 level measured? Using ELISA? When was the IL-6 level measured? 2 days after co-culture? Please describe.
5. In Figure 2B, what did the three different bars of each condition represent? Where is Condition 05?
6. Based on Figure 2C, the vGlut1 expression was very weak in Channel 2, but the quantitative data in Figure 2D showed that vGlut1 expression level was comparable between different channels. Please explain this discrepancy.
7. The MEA data in Figure 2E is one of the most important pieces of data in their study, but the data was very confusing. In Line 187, they stated “These results indicate an increase in nerve impulses when MoDCs are present in channel 1 of the device.” I think this is referring to condition 02, but the author only show the result of one experiment (since there were no error bars). How can they draw such a conclusion considering that the measurements in condition 01 showed such a great variation between replicates? Why is condition 03 missing in the figure? Also, the neurons in condition 02, 04 and 05 seemed not to be in a good condition because their firing rates were much lower than that in condition 01. The authors may consider showing IF of apoptotic markers with neuronal markers on their cells after MEA measurement to demonstrate that the neurons can still survive after 2 days of co-culture.
Minor concerns:
· Please give the full name of the terms in the first mention before using the abbreviations
· Some mis-labeling of figures in the main text.
Reviewer 3 Report
The authors of the “Human Chip model to study gut-brain axis neuro-immunological communication involved in Inflammatory Bowel Diseases.” have written well, but there are some comments that need to be addressed:
1. There are a few grammatical errors and phrasings in this manuscript that need to be corrected.
2. The figures are of poor quality. For example, the axes of Figure 1D are invisible to any reader.
3. Figure 2 is of poor quality, especially 2B and 2C.
4. Figure 2C has no meaning. The author should annotate based on the channel or whatever it means in the figure and relate it to their discussion.
